

# GC Insights: Identifying conditions that sculpted bedforms - Human insights to build an effective AI

John K. Hillier[1], Chris Unsworth[2], Luke De Clerk[3], Sergey Savel'ev[3]

[1]Geography and Environment, Loughborough University, Loughborough, LE1 3TU, UK
[2]School of Ocean Sciences, Bangor University, Bangor, LL59 5AB, UK.
[3]Dept. Physics, Loughborough University, Loughborough, LE1 3TU, UK.

*Correspondence to*: John K Hillier (j.hillier@lboro.ac.uk)

**Abstract.** 42 survey participants demonstrate that it is visually possible to recognise the type of flow that created bedforms (e.g. sand dunes, riverbed ripples) from short distance-depth profiles, but this is much harder for individual forms. An
interpreter's geoscience expertise does not help, indicating a machine learning or 'AI' algorithm might be trained well from the data alone, especially if multiple bedforms are used.

## 1 Introduction

Environmental flows shape the surface they flow over. The variety of features produced (e.g. sand ripples on a beach), known as bedforms, reflect and preserve characteristics (e.g. speed, depth) of the flowing ice, water or air (Venditti, 2012; Bullard et
al., 2011; Storrar and Stokes, 2007). The relationships between bedform morphology and flow are contested where observation is extremely difficult, such as under ice-sheets (e.g. Clark et al., 2018; Hillier et al., 2018; Rose, 1987; King et al., 2009), and best understood for unidirectional water flow over sand in a laboratory setting, mimicking a river (e.g. Fig. 1a).  Even in this idealised fluvial setting, it is difficult to construct a 1-to-1 link between bedform type (e.g. ripples or dunes) and specific flow conditions (Venditti, 2012; Froehlich, 2020). Illustratively, ripples have a higher aspect ratio ($H/L$, for height ($H$) and length
($L$)) than dunes (e.g. Allen, 1968); yet the observational ranges overlap (Venditti, 2012; Yalin, 1972), creating uncertainty when attempting to link morphology with hydraulic conditions. Many variables related to hydraulics and/or the physics of sediment movement have been proposed to remove the overlap in bedform stability diagrams such as Fig. 1a (see Venditti, 2012). Only recently, has a distinct and non-overlapping zonation of bedform type and flow structure been developed using a quantity called shear velocity (Duran Vinet et al., 2019). Inverting this result may help realise the aspiration of developing a
means to reliably infer flow conditions from bedform morphology (e.g. Duran Vinet et al., 2019; Venditti, 2012; Myrow et al., 2018), which is often the only option (e.g. sedimentary structures preserving the geological past, Mars)(e.g. Ohata et al., 2017; Edgett and Lancaster, 1993).

Machine learning or 'AI' algorithms, such as Artificial Neural Networks (ANNs) offer an opportunity to examine this problem
as they do not assume simple (e.g. linear or 1-to-1) relationships between inputs and predicted variables (Wang et al., 2009;





Faruk, 2010). This has been attempted for experimental parameters (Froehlich, 2020), but not for bedform morphology. There may be unexploited quantitative morphological subtleties to categorise bedforms, or even to accurately position them on stability diagrams. This work examines the scope for using ANNs to distinguish the flow conditions in which relict bedforms originated by asking if the ability exists in non-artificial (i.e. human) intelligence for two particulars:


> **Q1** - Is it possible to identify the environment (e.g. river, desert) of a bedform's genesis from its shape?
>
> **Q2** - In the fluvial environment, is it possible to distinguish flow conditions?

## 2 Method, Data & Ethics

An online survey (Supplementary Material) was conducted, initially at the '*Non-equilibrium flows and landforms*' workshop
(19[th] May 2021), with participation expanded using authors' close contacts (friends, colleagues, and family). For Q1, participants attributed distance-depth profiles across 34 individual bedforms, and 13 bedform sequences (≥3 bedforms) to one of four environments (*fluvial* [river], *glacial*, *marine*, *aeolian* [desert]). For Q2, participants ranked three profiles according to flow strength (shear velocity), thrice for individual forms, and thrice for bedform sequences. Examples were provided to isolate visual shape analysis from prior knowledge (Fig. 1b), black and white profiles were used to exclude contextual clues (e.g.
dataset characteristics, other features in the landscape), and the order of options (e.g. B, A, C) was shuffled for each participant to prevent bias. Scale (e.g. metres) readily distinguishes environment without using bedform shape, so it was not given.

Aeolian data are ASTER (v2) across linear and transverse dune types from the Namib desert (Bullard et al., 2011), glacial are from near Lough Gara in Ireland (Hillier and Smith, 2008), fluvial are from four laboratory experiments (Expts. 1-4) of
increasing shear velocity (Unsworth, 2015), and marine data are from the Irish Sea. Representative examples of individual bedforms and sequences were manually selected from these datasets.

Ethical approval was given by the Ethics Review Sub-Committee at Loughborough University.

ANN analysis to follow up the survey used a Multi-Layered Perceptron (MLP) with four hidden layers with 28, 56, 56 and 28 nodes, each with a ReLU activation function. Height (*H*) and width (*W*) fitted using the SWT algorithm (Hillier, 2008) and a frustum approximation (Hillier, 2006), see Fig. 1c & d, were input to predict the flow regime (experiment number). Groups consisted of five bedforms. Selection was random, and without replacement from a single time series until all bedforms in that time series were used. Weights and biases were updated using the Adam Optimiser of PyTorch using a loss function that
calculates the Mean Squared Error (MSE), all within a feedforward back-propagation approach.





## 3 Results

Of the 42 survey participants 25 self-identified as geoscientists, and 16 did not. For Q1, participants correctly identified the one of four environments (e.g. fluvial, aeolian) in which individual features originated 32% of the time, slightly if significantly (2-tailed t-test, $p \ll 0.01$) better than the 25% expected of guesswork. This rises to 51% for bedform sequences. For Q2, participants ranked entirely correctly 3 flow strengths (Expts. 1-3) for 46% of individual features, and 60% of sequences, much better than the 16% expected of guesswork (conservatively assuming no strength is repeated, $p \ll 0.01$).

In none of the questions did geoscientists perform better than non-geoscientists, with mean percentages of correct answers being indistinguishable (2-tailed t-test, $p > 0.05$). The overall sentiment is encapsulated by one comment:

> "*I felt this was a geometrical exercise of recognising same patterns at different scales. I did not feel that my experience as an "expert" in bedforms really made any difference from, say, my son taking the test.*"

## 4 Discussion

The survey results clearly demonstrate it is possible to distinguish fluvial flow conditions from distance-depth data of the bed, and that an ANN should perform better if utilising sequences of bedforms rather than evaluating individuals in isolation. Geoscientists' *a priori* and contextual knowledge added little, indicating that training an ANN on these data alone should be productive.

Morphologies from differing environments (e.g. glacial, fluvial) are often viewed as similar, indicators of analogous processes at work (e.g. Shaw, 1983), and modelled with identical equations (e.g. Fowler, 2002; Duran Vinet et al., 2019) or statistics (e.g. Hillier et al., 2016; Einstein, 1937). Several participants commented that their ability to distinguish environments might be to do with characteristics of the data (e.g. smoothness due to data resolution), not bedform shape, highlighting a potential pit-fall of training an ANN on raw distance-depth data. Another limitation is that ANNs performing pure pattern recognition need 1000s of training datasets (e.g. Bishop, 1996), which are not readily available in geoscience, and are apparently not needed by the survey's participants. Potentially, this requirement can be avoided by pre-processing to identify key aspects of a geomorphological pattern (e.g. Shumack et al., 2020).

Shear velocity increases non-linearly across experiments 1 to 4. Fig. 1c-e present the outputs of a follow-up analysis into (i) the plausibility of building an ANN using only 4 time-series if pre-processing heights by fitting flat-topped cones to bedforms (Hillier, 2006, 2008), and (ii) the potential benefit of using short bedform sequences. Visually (Fig. 1d), the individual shapes (*H, W*) overlap between experiments, but the trends and averages over a number of points are distinctly different. This maps





directly to results of the ANN (Fig. 1e). Individual forms are weakly predicted (light grey, $r^2 = 0.11$), but sub-sets of 5 bedforms more strongly so (grey, $r^2 = 0.56$), particularly if very small bedforms present in all experiments ($H < 0.5$ cm) are excluded

(dark grey, $r^2 = 0.80$). Interestingly, simply taking the first four moments (mean, variance, skew, kurtosis) for arbitrarily located segments of the time-series, each 100 seconds long, yields better prediction ($r^2 = 0.89$). Thus, insights from the participants have contributed to building an effective AI to reliably infer flow conditions from bedform morphology, yet a trade-off may exist between the most discriminating ANN (e.g. from statistical properties - (e.g. Malinverno, 1988; Powell et al., 2016; Singh et al., 2011)) and ease of relating outputs to process understanding and traditional shape parameters (i.e. $H$ and $W$). In future,

in transitional, non-equilibrium conditions (e.g. see Myrow et al., 2018), ANNs may be key to disentangling forms and flow.

**5 Audiences and wider application**

Geoscience communication in this work was by engaging with non-geoscientists as an integral part of a research project. In doing so, it introduced geomorphology as a discipline and illustrated an active research question. The practical geoscientific

insight of wider interest is that derived properties of landscapes (e.g. shape parameters for several dunes) can assist training an AI where data are limited (i.e. observations of Earth, physical experiments).

**Acknowledgements**

The "Non - Equilibrium Turbulence and landforms workshop" was organised by Tim Marjoribanks, Chris Keylock, Christopher A. Unsworth, Daniel R. Parsons and Jonny Higham, with support from the British Society of Geomorphology. We thank Matt Baddock for preparing the aeolian data, and the 42 anonymous participants for completing the survey.

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

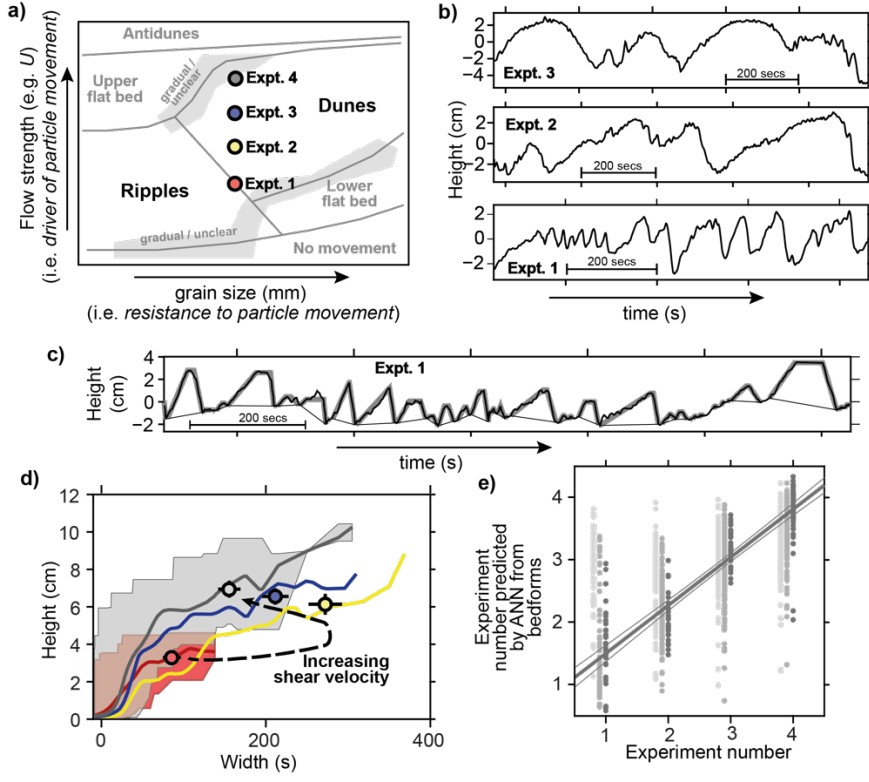

**Figure 1: (a) Illustrative bedform stability 'phase diagram' for unidirectional fluvial (i.e. river) bedforms, synthesized from multiple sources (Ohata et al., 2017; Lewis and McConchie, 1994; Southard and Boguchwal, 1990). Main types considered here (i.e. ripples and dunes) are highlighted. Experiments 1-4 are positioned indicatively. (b) Time-series like those given unannotated to participants, i.e. one from each experiment 1-3, all scaled to the same dimensions. Horizontal axis is time because in the flume tank a stationary sensor recorded height as bedforms passed beneath it. (c) Example of how $H$ and $W$ are determined. Measured heights (thick black line) are processed using the SWT algorithm to identify bedforms, drawing a line beneath them (thin black line) then approximated as flat-topped cones (grey lines). SWT parameters as Hillier (2008). (d) Height-width relationships for the 4 experiments, with colours as in (a): lines are sliding means with $W$ (Gaussian weights, width 60), shaded areas are full ranges for Expts. 1 & 4, and dots are the means ($\pm 2\sigma$) of upper quartile of the data when the small bedforms (i.e. $H < 0.5$ cm) are excluded. (e) Comparison of actual experiment number for out of sample prediction by the ANN using $H$ and $W$: individual bedforms (light grey), subsets of 5 bedforms with (grey) and excluding (dark grey) small bedforms.**