# Peer review of "Identifying conditions that sculpted bedforms: Human insights to build an effective AI"

_Geoscience Communication, 2021_

## Author Comment (AC2)

Replies to comments on **gc-2021-22 "** *GC Insights: Identifying conditions that sculpted bedforms – Human insights to build an effective AI*"

Comments were kindly provided by two reviewers (RC1, RC2). Both reviewers highlighted the challenge of clearly communicating within this concise format, and we have made numerous modifications to address this.

Please find below our response to the comments. Comments are in grey, and responses in black. Although a fully revised manuscript is not yet prepared, we use 'changed' to indicate some simpler modifications where it was easiest to communicate by simply actioning the comment (see provisionally revised manuscript at the end of this pdf).

**RC1**

This paper explores an interesting idea. I particularly liked the idea of comparing and complementing AI-based and human decision-making, and the insights concerning the relevance of expertise are interesting.

As a reader with a background in machine learning and AI, rather than geoscience, I have a number of comments and questions concerning the current version of the manuscript:

- It is not clear to me what precisely the input to the ANN was, or what the task was (classification of time series of photos, I think, but this is left rather implicit). Was the input a sequence of images in each case? How many were included in the training / testing sets? How was the algorithm's performance evaluated? In general, there is not sufficient information in the paper to understand exactly what was done in the neural network part of the study. I would like to see complete detail, and/or reference to an implementation available for scrutiny/study.

> In order to provide sufficient information to reproduce the work, and yet comply with the length-limit of the *GC Insights* format, the input data and pseudo-code for the ANN are now provided as Supporting material. The algorithms performance was ultimately evaluated using out-of-sample prediction of experiment number (Fig. 1e) using 40% of the data withheld from training/validation. Performance during ANN training was using RMSE, with details of this now in the pseudo-code provided.

- Why the particular choice of ANN? Why those numbers of neurons and in that configuration? Were other options considered?

> Based on our prior experience, and the limited data available, we selected a small network (i.e. limited number of layers and nodes). After this, there was some *ad hoc* refinement, leading to the configuration used. We do not claim that this is optimal, and now explicitly state (e.g. in the Abstract) that this is a preliminary analysis. This is in line with the guidance for a *GC Insights* article that work must "*be well-founded and methodologically robust, based on evidence or analysis that can be openly inspected, but does not have to be comprehensively explored*".

- It seems that the results of the ANN study are very preliminary, and no conclusions can yet be drawn. (The manuscript uses phrases like "that training an ANN on these data alone should be productive".) Is that right? What are the conclusions drawn from the ANN work to date, or is it still at the speculative stage? The paper was unclear, to me, about this.

> The results of the ANN study are indeed preliminary; their purpose is to demonstrate that avenues suggested by the survey might be fruitful.  The abstract and text have been re-phrased to be clearer that this is the intention. To use the example cited, we have clarified in paragraph 1 of the Discussion that the 'should' comes from the survey and in paragraph 2 stated that the ANN work provides initial numerical support for the idea. So, conclusions from the ANN are

1. It is possible to build an ANN with some predictive power (i.e. one successful example is sufficient to demonstrate the principle).
2. For this, sequences of bedforms produce better results than individual ones
3. And 'help' in the form of pre-computing morphological parameters mitigates the issue of data volumes that are available in this geoscience context.

- The manuscript claims that "Thus, insights from the participants have contributed to building an effective AI to reliably infer flow conditions from bedform morphology". I didn't see justification for this in the paper. The argumentation and explanation of the ANN results (and the lack of clarity around which are speculations and which are results) make it difficult to see where this comes from.

> This was a typographic error. We intended to say 'will contribute' rather than 'have contributed', because demonstrating the 'have contributed' needs a successful result that is following in a full paper after building an AI incorporating the insights gained here and fully applying it. This sentence has been removed in the rewrite of the discussion, but we will endeavour to avoid similar in the revised manuscript.

I think the submission could make for an interesting contribution, if the method, results, and contributions were more clearly stated.

> Throughout, we have re-written the manuscript with the aim of more clearly describing our work. In particular, pseudo-code and data are now provided in Supplementary Material to clarify the method. The ANN results are now in the Results section, instead of being presented as a follow-up, and we have re-phrased to clarify the narrative of the paper and thus its contribution.

**RC2**

The paper is a concisely written report of a survey of a mix of geoscientists and non- geoscientists to determine whether bedforms from different processes and environments could be classified based on shape alone. The results are interpreted in the context of building an artificial neural network model for the same task. The survey results indicate that non-geoscientists perform as well as specialists at this task, as it is primarily a shape matching exercise that does not require specialist knowledge. Unsurprisingly, identifying environment from an individual bedform was more difficult for respondents than when a series of bedforms were provided. The paper then applies an ANN model to the task with qualitatively similar results. The involvement of the non-geoscientists in the survey group is the "geoscience communication" aspect of the paper.

> This is a fair summary. Although the results may be unsurprising, they were and continue to be of use to the PhD student and inter-disciplinary supervisory team in focussing the possible routes forward in designing an ANN. Illustratively, the use of derived parameters such as height ($H$) and width ($W$) is a 'natural' approach for a geomorphologist (e.g. lead author), but not to machine learning specialists, even in a data limited scenario. As such we argue that there is value in exploring and confirming what might appear obvious from some viewpoints.

The shorter-than-short article format seems to have resulted in a few leaps being made that would benefit from further explanation. I will highlight these in my technical comments below.

There are also a few scattered errata
> Thank you. We have endeavoured to find and correct these.
...., but overall the text is clear.
> Thank you.

Abstract, elsewhere throughout - the way the profiles are described is highly inconsistent. Here and elsewhere they are called distance-depth profiles, presumably referring to bathymetric depth, but they are plotted as height vs distance, ie, topographic profiles, and presented as such to the survey respondents. The text later switches to talking about the shear velocity experiment profiles as elevation time series, the explanation for which is not in the main text but buried in the figure caption. While the timeseries could be recast as spatial series related through the migration rate of the bedforms, this step seems to have been omitted, so they are really timeseries presented as topography to the survey respondents and lumped in with the other experiments in the paper abstract and introduction as topography.

> We agree that within the context of a concise paper, for a general geoscience communication audience (i.e. including non-geoscientists), a consistent approach to describing the profiles without using jargon (e.g. topography, time-series) is best. This simplification is now introduced explicitly, at the same point in the

manuscript as the data are introduced. Of course this hides the detail, some of which is retained in the figure caption (i.e. glacial and aeolian are from topographic DEMs, fluvial are time series of forms passing under a sensor, and marine data are time series of acoustic measurements taken from a moving boat).

Line 31 - "this has been attempted for experimental parameters" is vague, what experiments? What parameters? ML has been applied to lots and lots of things...
> Froehlich (2020) used machine learning to attempt to predict fluvial flow regime from the standard experimental parameters (e.g. Froude number) in numerous flume tank experiments and natural streams recorded in the literature. This is the closest ML application to what we are attempting, but the reviewer identifies that it is too distant to include without further explanation. This previous work is now incorporated in a more general statement in the sentence before, and the confusing statement removed.

46 - "Scale readily distinguishes..." I get that the purpose was to see if there was scale- independent data in the shapes that would allow discrimination of these bedforms, but if the ultimate purpose is to build a model to do so, wouldn't scale be included as it's such reliable indicator?
> An exciting possibility we are exploring is the creation of a scale-independent ANN model, which might have the capability to be used across environments as from this it might be possible to gain insights into any unifying elements or characteristics of physical processes (e.g. to comment on Martian dunes from Earth analogues). For this, our current view is that scale is best omitted else the ANN would in effect be internally creating a separate model for each environment without transferability.

46 - Also, it appears that some sense of relative scale WAS in fact provided on the survey. The tick mark spacing on the frame of each profile seems to have been scaled along with the profiles when the extracted bedforms were rescaled. It is therefore easy to tell from the height of the bedform relative to the tick mark spacing that certain profiles came from a specific "training" profile. It would be hard to determine whether survey respondents registered this or not post hoc, and to me it calls the results into question. Why were the tick marks included on the "scale free" profiles at all?
> The reviewer is correct that, in hindsight, it would probably have been better to omit tick-marks entirely from the survey. However, we are confident the results we focus upon are sound as we asked questions in the survey to elicit whether or not such effects/biases were present. Our view for this primarily comes from two questions. First, 'a scale' (or similar) was the dominant response in a number of questions during the survey (e.g. Q3.36) paraphrased as "*What contextual information would have helped*?". Thus, if participants used tick-marks as a reliable scale indicator, it was a subconscious process. Also, when asked if anything made them feel confident of otherwise about their decisions (e.g. Q3.35), no participant mentioned using the tick-marks in the way the reviewer describes.
> The survey questions and responses are provided in the accompanying material for inspection, however to be concise and in line with the scope of the *GC Insights* format (i.e. "*does not need to be fully explored*") various aspects of the contextual information there (e.g. data resolution, roughness) are not explicitly considered in the main text. Consistent data quality, however, is one reason for our focus on the fluvial experiments in terms of our analysis and conclusion.
> We also focus on conclusions that are insensitive to such possible weaknesses in the research method e.g. '*short sequences are better than individuals*'.

57 - the parenthetical "(experiment number)" is initially confusing, replace perhaps with "(coded by experiment number) "or something like that.
> Text modified in line with the reviewer's suggestion.

58 - here is where the text starts talking about time series instead of topography with no explanation unless you happen to read the figure caption first.
> Thank you for this guidance on first uses of terminology, we now use 'distance-height profiles' throughout unless unavoidable for technical correctness.

63-65 - "For Q1... expected of guesswork" this sentence is a bit overloaded with info and convoluted to read.

65-67 - "For Q2..." this sentence is also convoluted.
> We note the reviewer's point, however the GC Insights format dictates concise wording. The sentences may need re-reading, but we believe they are clear, saving words for descriptive and less technical parts of the article. We have removed the technical point in brackets relating to the 16% expectation to avoid confusion.

81 - more explanation needed, as far as I know glacier bed topography and aeolian ripples are most certainly NOT modeled by the same physical equations... unless you mean not "modeled" but "described empirically"... there are process and form similarities across many environments, but perhaps this statement is oversimplified?
> Thank you for reminding us that bedforms (e.g. aeolian, glacial) are often modelled with different equations, and seen as different, but they are similar enough that they *can be* described (quantitatively, qualitatively, using equations) similarly. To soften our statement, therefore, the text has been modified to change 'are often' to 'can be'. The references used all take a view where the environments are modelled similarly e.g Fowler (2002) in "*Evolution equations for dunes and drumlins*" uses the same Exner equations for both.

90 - this is hard to follow. There is a leap between the previous mention of preprocessing to the description of the preprocessing method, which is embedded in a topic sentence.
> Pre-processing is now introduced in the Methods section.

83-95 - This section of the discussion justifies the preprocessing and introduces the ANN, is it really discussion? Seems more like methods.
> Thank you for this prompt. We agree. The initial structure is a legacy of the way in which the work unfolded. The original mini-project reported here was only the survey, and we added the preliminary follow-up ANN work later, which in reality aided our discussion and use of the survey results. The ANN is now introduced in the methods section, with technical detail now supplied in accompanying pseudo-code, and results of the ANN are now in the results section.
How the preprocessing method follows from the survey results needs more explanation.
> Please see modified discussion.
As for the potential pitfall of training the ANN on the raw data, was this attempted? With what result? It is merely asserted that it might be a problem.
> Thank you. Our results on running the ANN on the raw time-series data are now reported, allowing us to more clearly present this argument.
As for the fitting of flat topped-cones, there are different ways to fit things, which may depend on scale of interest etc.... More info about this step is needed. (Actually some detail is provided back in the methods section - but it's not clear that the SWT/frustum algorithm described there is the same as the "fitting flat topped cones" described here. Also, SWT is never defined as an acronym.)
> Thank you for this comment. We entirely agree that fitting the cones, or indeed other methods of deriving shape parameters, will produce somewhat different results. The method used is scale-independent (i.e. fitting is after scaling into a unit box), and explicitly accounts for data gaps and density variations. Full details are given in the references cited (Hillier, 2006; Hillier, 2008), sufficient to reproduce the method, and identical parameters (e.g. for the SWT filter) are used here as in those papers.
> The pre-processing is now introduced in Methods, and the term 'frustum' no longer used to clarify that flat-topped cones are being fitted by the SWT/shape-fitting algorithms.
> SWT acronym is now defined at first use.

95 - The aside here about taking the statistical moments of the signal seems to obviate the other preprocessing. It's also not clear whether it means that the ANN is more successful when any of these moments are used individually or if they are used together. The next sentence, the conclusion that insights from the participants led to a more successful model, does not seem to follow from this sentence about statistics.
> For clarity and focus this sentence about statistical moments has been removed. The reviewer rightly identifies that this sentence is somewhat tangential, and in a concise paper it is necessary to focus on key illustrative results of work done.
> There are issues with the use of these four statistical moments together. For example, much of the skill comes from mean height, which is arbitrary with respect to bedform processes and so very unlikely to remain

useful even when transferring to different fluvial experiments. Unpacking this is beyond the scope of a *GC Insights* article. It will be covered in the geomorphology / machine learning paper to follow.

99 - "in the future"
> Text modified.

So, I'm not really sure what to make of this study. It seems like a good exercise overall, to understand whether expert knowledge might help build a better ANN classifier, but most of the insights could have been gleaned just from trying the ANN in the first place.
> We fully accept the comment about just trying various ANNs, but after trying various routes, and building and training numerous ANNs with relatively limited success, the survey helped up to focus the design process (i.e. provided insights to help design an effective ANN). Perhaps the main insights are (i) it is possible and (ii) there is value in 'helping' the ANN in the context of limited data, which is the opposite of machine learning communities default desire to train ANNs on raw data so as to impose as few *a priori* assumptions as possible. The text has been adapted to more clearly communicate this.
Coupled with some of the issues of scale info on the survey, the unclear link between the survey results and how they informed building the ANN, and the condensed article format leading to omitted context, it's hard to recommend publication without substantial revision.
> Thank you for your constructive feedback, we have revised the text to clarify how the survey results informed the building of the ANN, and focussed the text to clarify its purpose.

**Identifying conditions that sculpted bedforms: Human insights to build an effective AI**

John K. Hillier[1], Chris Unsworth[2], Luke De Clerk[3], Sergey Savel'ev[3]

[1]Geography and Environment, Loughborough University, Loughborough, LE1 3TU, UK
5  [2]School of Ocean Sciences, Bangor University, Bangor, LL59 5AB, UK.
[3]Dept. Physics, Loughborough University, Loughborough, LE1 3TU, UK.

*Correspondence to*: John K Hillier (j.hillier@lboro.ac.uk)

**Abstract.** In a survey, 42 participants could visually recognise which flow conditions created bedforms (e.g. sand dunes, riverbed ripples) from distance-height profiles, particularly if multiple bedforms are present. Furthermore, an interpreter's geoscience expertise does not help whilst only one set of 'training' examples are needed, together indicating that an effective machine learning or 'AI' algorithm might be trained successfully from limited data alone especially if it is 'helped' (e.g. by pre-processing to fit flat-topped cones). 
[revised manuscript text omitted]

---

## Author Response (AR2)

Response to comments on **gc-2021-22-R1 "** *GC Insights: Identifying conditions that sculpted bedforms – Human insights to build an effective AI*"

Comments were again kindly provided by two reviewers (RC1, RC2), and we are glad that they both felt the changes in Revision #1 greatly improved the manuscript. Please find below our point-by-point response to this second round of comments. Comments are in grey, and responses in black. A manuscript with changes tracked is also provided. We have also made some minor changes to remain within the word limit.

We hope that the editor will find the manuscript suitable for publication after these minor corrections.

All the best

John
(on behalf of the authors)

**RC1 (Anonymous)**

I would like to thank the authors for the detailed point-by-point response and revision, which was useful to better understand the work. The clarity of the article has substantially improved, and the addition of the supplementary material is valuable.
> Thank you.

I have two remaining comments, one technical and one more conceptual:

- In Fig 1(e), what is the meaning of the diagonal line linking the datasets? My understanding is that these are from different experiments, and are only ordered indicatively. Is there some trend that can be meaningfully arrived at here, given the qualitative differences between the experiments? If not, the line may be inappropriate. (There is some indication of 'non-linearly increasing velocity', but it perhaps if the authors want to draw out a relationship, that should be explored more.)
> The reviewer correctly identifies that the data are from four different experiments. These are ordered in terms of shear velocity, which increases monotonically yet non-linearly from Expt. 1 to Expt. 4. As such the ordering is useful, yet we accept that adding a trend line is not strictly statistically justified. It was added for visual and illustrative purposes only, and we have therefore removed it as upon reflection the improvement in the methods is visible without it.

- Underlying the discussion is, I think, the idea that ML techniques might work well where non-experts have been demonstrated to make good decisions based on particular visual data. This is a really interesting idea, but I wonder if it could a) be made more explicit, and b) if this comes from somewhere in the literature or is the supposition of the authors. Either way, drawing out this as a testable proposition, even briefly, would strengthen the paper, I think, as it has implications beyond just this domain.
> We thank the reviewer for focussing us on this idea, and requesting that we clarify our presentation of it. It is a speculation of the authors, based on our observation of the results of the human participants, and we have endeavoured to draw this out briefly as a testable proposition. This interesting idea is now in the final paragraph of the paper, giving it potential to be recognised outside the immediate domain of this study.

In general, the idea of establishing the sufficiency of a particular set of data inputs (e.g. shear velocity), to be able to determine an outcome, seems to be an interesting and valuable contribution.
> Thank you.

Reviewer #2 (Ward)

The revised version of this paper is better-organized and focused, with a more consistent terminology, and has addressed most of the review comments from the previous round. I think it can be published following minor revisions.

> Thank you.

It still appears as if there are some small gaps in the information provided from paragraph to paragraph. This is much improved from the first version, and I appreciate the difficulty imposed by the extra-short article format. I hope the suggestions below can be accommodated without too much added length.

> Please see detailed responses below.

Abstract - The abstract needs rewriting. It currently begins with a result, followed by a lengthy and somewhat confusing sentence that jumps topics from survey participants to machine learning, with a final inference about pre-processing. It is not clear how this inference follows from the beginning of the sentence. I suggest starting with something like the final sentence, modified: "Preliminary investigations with an ANN illustrate that a geoscience comm activity…", and split the middle sentence to clarify each clause.

> We have re-written the abstract, starting by placing the sentence that the reviewer suggested at the beginning.

39-40 Should the question of whether expert knowledge is useful be added here? As Q1a perhaps? It leads to one of the more interesting outcomes, which is reported in the abstract and section 3, but is never introduced as part of the study in the current version.

> The inclusion of both experts and non-experts is now made explicit at the start of Section 2 (Methods). Unfortunately, like the question of single bedforms vs sequences of bedforms, there is no space available to explicitly introduce this explicitly as a separate question.

56-57 re: time and distance profiles. While the presentation of this disparity is much improved, this will still be a bit confusing to the uninitiated.

Suggestion: could you simply elucidate, e.g. "…referred to as distance-height profiles, although the fluvial experimental data were collected as time series of bedform height passing below a sensor with time as a proxy for distance"? Pressed for space, you could find a few words at the beginning of the sentence and just say "For simplicity, these varied data…"

> As suggested, the start of the sentence has been shortened to save words, and we state that time is used as a proxy for distance in the fluvial data. We also now point to reader to the caption of Fig, 1 where there is a brief explanation of how the data are collected that corresponds well to the reviewer's suggestion (i.e. "*Distance-height profiles (strictly speaking time-series) like those given unannotated to participants, i.e. one from each experiment 1-3, all scaled to the same dimensions. Horizontal axis is time because in the flume tank a stationary sensor recorded height as bedforms passed beneath it.*")

Note from a physical standpoint I am not convinced the height-time and height distance profiles are directly equivalent, because bedform sizes vary in these profiles, and don't different sized bedforms migrate at different rates? However I don't think it will affect the outcomes presented here.

> From a physical standpoint, we agree. Different bedforms do indeed migrate at different rates, but this does not affect the conclusions presented here.

85-89 There seems to be a gap here, the paragraph starts talking about the baseline ANN experiment (which didn't work so well) and then does not signal that the remaining results are from the ANN with shape-fitting. Or is the difference the ANN with multiple bedforms? Suggest clarifying which results correspond to which ANN experiments. This will make the following discussion (101-107) more impactful.

> We have included a sentence explicitly clarifying that all experiments after the baseline one are based upon shape-fitting inputs to the ANN.

Again I appreciate the difficulty of the short format, the authors have done an admirable job of condensing a couple of potentially complex topics, so I hope it is helpful to point out where an outside reader will find a leap or two being made.
> Thank you.

Dylan Ward